# LncRNA and Protein Expression Profiles Reveal Heart Adaptation to High-Altitude Hypoxia in Tibetan Sheep

**DOI:** 10.3390/ijms25010385

**Published:** 2023-12-27

**Authors:** Zhaohua He, Shaobin Li, Fangfang Zhao, Hongxian Sun, Jiang Hu, Jiqing Wang, Xiu Liu, Mingna Li, Zhidong Zhao, Yuzhu Luo

**Affiliations:** Gansu Key Laboratory of Herbivorous Animal Biotechnology, Faculty of Animal Science and Technology, Gansu Agricultural University, Lanzhou 730070, China; hezh@st.gsau.edu.cn (Z.H.); zhaofangfang@gsau.edu.cn (F.Z.); sunhx@st.gsau.edu.cn (H.S.); huj@gsau.edu.cn (J.H.); wangjq@gsau.edu.cn (J.W.); liuxiu@gsau.edu.cn (X.L.); limn@gsau.edu.cn (M.L.); zhaozd@gsau.edu.cn (Z.Z.)

**Keywords:** Tibetan sheep, heart, lncRNA, proteomic, high-altitude hypoxia

## Abstract

The Tibetan sheep has an intricate mechanism of adaptation to low oxygen levels, which is influenced by both genetic and environmental factors. The heart plays a crucial role in the adaptation of Tibetan sheep to hypoxia. In the present study, we utilized transcriptomic and proteomic technologies to comprehensively analyze and identify the long non-coding RNAs (lncRNAs), genes, proteins, pathways, and gene ontology (GO) terms associated with hypoxic adaptation in Tibetan sheep at three different altitudes (2500 m, 3500 m, and 4500 m). By integrating the differentially expressed (DE) lncRNA target genes, differentially expressed proteins (DEPs), and differentially expressed genes (DEGs), we were able to identify and characterize the mechanisms underlying hypoxic adaptation in Tibetan sheep. Through this integration, we identified 41 shared genes/proteins, and functional enrichment analyses revealed their close association with lipid metabolism, glycolysis/gluconeogenesis, and angiogenesis. Additionally, significant enrichment was observed in important pathways such as the PPAR signaling pathway, glycolysis/gluconeogenesis, the oxoacid metabolic process, and angiogenesis. Furthermore, the co-expression network of lncRNAs and mRNAs demonstrated that lncRNAs (MSTRG.4748.1, ENSOART00020025894, and ENSOART00020036371) may play a pivotal role in the adaptation of Tibetan sheep to the hypoxic conditions of the plateau. In conclusion, this study expands the existing database of lncRNAs and proteins in Tibetan sheep, and these findings may serve as a reference for the prevention of altitude sickness in humans.

## 1. Introduction

The Tibetan Plateau is known as the “Roof of the World” and the “Third Pole”, with an average altitude of more than 4000 m, accounting for about a quarter of China’s land area [1,2]. It has been found that ambient partial pressure of oxygen (PO_2_) decreases with altitude and that PO_2_ at 4000 m above sea level is only about 60% of that at sea level [3]. At high altitude, the lower PO_2_ is more likely to lead to hypoxia, which in turn affects the reproduction and survival of animals. However, after a long period of evolutionary selection by nature, a number of animals have emerged on the Tibetan Plateau that are able to adapt to the low-oxygen and high-cold environment, such as wild yak (*Bos mutus*) [4], Tibetan antelope [5], Tibetan sheep, etc. As one of the three original sheep breeds in China, Tibetan sheep are mainly distributed in Tibet, Qinghai, and other areas at an altitude of 2500–5000 m, and the number of Tibetan sheep is huge [6,7] due to the wide distribution range of Tibetan sheep and the altitude differences between different regions. Therefore, the study of the adaptation mechanism of Tibetan sheep to the low oxygen environment at different altitudes could help to discover key information about the adaptation of other plains animals to the highland hypoxia environment.

The mechanisms of altitude sickness are complex. Hypoxia is the cause, while upper respiratory tract infections, fatigue, cold, stress, hunger, pregnancy, and other factors are all causative factors [8]. Hypoxia triggers the consistent activation of HIF, which in turn triggers the activation of numerous hypoxia-related genes, including EPO, which facilitates the production of red blood cells. This process becomes excessively active during acute hypoxic exposure and can result in abnormal erythropoiesis. Consequently, the elevated number of red blood cells increases blood viscosity, predisposes individuals to pulmonary hypertension, and causes damage to the microcirculation [3,9].

To a certain extent, the strength of cardiopulmonary function could reflect the ability of animals to adapt to the highland hypoxia environment. When confronted with a hypoxic environment, the animal organism ensures adequate oxygen supply mainly by increasing pulmonary ventilation [10] and blood flow [11,12]. However, increased blood flow can lead to increased blood pressure, which may lead to the development of pulmonary hypertension [13]. Compared with plain animals, indigenous animals at high altitudes have stronger cardiopulmonary function, mainly reflected in the following aspects: higher resting respiratory rate and alveolar ventilation efficiency, such as Tibetan sheep and Peromyscus maniculatus [14,15]; larger cardiopulmonary volume, such as plateau pigs [16] and yaks [17,18]; and thicker cardiac wall and epicardium, as well as richer intermuscular blood vessels, such as Tibetan balsam pigs [19]. In addition, animals at higher altitudes have higher red blood cell counts and hemoglobin content compared to those at lower altitudes, resulting in improved oxygen-binding transport efficiency [20]. In conclusion, as altitude increases, the internal organs and blood physiological and biochemical indexes of animals will undergo corresponding adaptive changes.

At present, transcriptomic and proteomic technologies are well-developed and widely utilized in the investigation of regulatory mechanisms of various economic traits in plants and animals. From a broader perspective, transcriptomics encompasses four main types of information: mRNAs, microRNAs, lncRNAs, and circRNAs [21]. Research has demonstrated that noncoding RNAs (ncRNAs) could also participate in and regulate genetic information through various means [22]. Long noncoding RNAs (lncRNAs) are a category of noncoding RNAs that are longer than 200 nt and transcribed by RNA polymerase II. They could be involved in different types of immune regulation through transcriptional and post-transcriptional mechanisms [23]. LncRNAs have been discovered to play significant roles in the developmental regulation of various tissues and organs, such as muscle [24], liver [25], heart [26], and lung [27]. Additionally, proteomics has been extensively employed in various biological studies, including the analysis of plasma proteomes associated with immunity [28], muscle proteomes associated with muscle development and fat deposition [29], and cardiac proteomes for hypoxic adaptation [30]. The transmission of molecular genetic information primarily occurs from deoxyribonucleic acid (DNA) to proteins through messenger ribonucleic acid (mRNA). However, selective gene expression involves numerous intricate interactions, and not all transcribed genes are translated into proteins [31]. Therefore, integrating transcriptomic and proteomic approaches may be more advantageous in uncovering certain complex biological genetic information.

In this research, we chose three heart tissues from Tibetan sheep at varying altitudes and employed RNA-Seqs and label-free proteomics to examine and analyze the expression patterns of differentially expressed lncRNAs and DEPs. By integrating the multi-omics data, we aim to identify the crucial lncRNAs, genes, and proteins linked to the adaptation to high-altitude hypoxia in Tibetan sheep, as well as explore potential connections among them. The findings of this investigation will contribute to a better understanding of the mechanism underlying the adaptation to high-altitude hypoxia in Tibetan sheep.

## 2. Results

### 2.1. Quality Analysis of Transcriptomic and Proteomics Analysis

During the transcriptome analysis, Tibetan sheep at three altitudes yielded 73,744,200, 72,917,734, and 89,686,072 raw reads, respectively. After removing splices and low-quality sequences, the three groups contained 73,324,286, 72,526,068, and 89,184,858 high-quality reads, respectively, which were suitable for subsequent analysis. The remaining reads accounted for approximately 99.43% of the raw reads (Table 1). Additionally, we discovered a total of 1071 novel lncRNAs in this study (Appendix A). Based on their relative positions in protein-coding genes, these lncRNAs could be classified into five main categories: sense lncRNAs (240), antisense lncRNAs (138), intronic lncRNAs (5), bidirectional lncRNAs (118), and intergenic lncRNAs (3763), with intergenic lncRNAs being the most abundant (Appendix A). Moreover, the correlation between the lncRNA and mRNA expression profiles of the measured samples was above 0.950 in this study (Appendix A).

To investigate lncRNAs associated with acclimatization to plateau hypoxia, we conducted a comparative analysis of lncRNA expression levels in the cardiac tissues of Tibetan sheep at varying altitudes. Our findings revealed 46 DE lncRNAs in the comparison between TS25 and TS35, 111 DE lncRNAs in the comparison between TS35 and TS45, and 91 DE lncRNAs in the comparison between TS25 and TS45 (Figure 1A,B, Appendix A). Furthermore, we identified two co-expressed DE lncRNAs among the three altitude comparison groups (Figure 1A). The clustering heatmap analysis of DE lncRNAs across different comparison groups demonstrated noticeable differences (Figure 1C).

In the proteomic analysis, a total of 50,264 precursors, 43,937 peptides, 4459 protein groups, and 4726 proteins were identified in all samples from the three altitudes. Furthermore, we detected 4717, 4715, and 4707 proteins in cardiac tissues from the three altitudes of TS25, TS35, and TS45, respectively (Appendix A). The statistical analysis of the number of protein peptide bars revealed that one-third of the total number of proteins (1273) had peptide bars above 11, while the remaining proteins had peptide bars below 11 (Appendix A). In the comparison groups of TS25 and TS35, TS25 and TS45, and TS35 and TS45, we identified 173, 193, and 223 differentially expressed proteins (DEPs), respectively. Among these, the numbers of up-regulated DEPs were 106, 84, and 88, respectively, while the numbers of down-regulated DEPs were 67, 109, and 135, respectively (Figure 2A–C, Appendix A). Additionally, we determined the number of DEPs that were common and unique in the three comparison groups at different altitudes and found that 22 DEPs were co-expressed (Figure 2D).

### 2.2. Functional Assessment of DE LncRNAs in Tibetan Sheep at Different Altitudes

To investigate the role of long non-coding RNAs (lncRNAs) in the adaptation of Tibetan sheep cardiac tissues to plateau hypoxia, we conducted prediction analyses to identify target genes of lncRNAs. We found 5 potential target genes from cis-acting lncRNAs and 279 potential target genes from trans-acting lncRNAs. Subsequently, we performed KEGG and GO enrichment analyses on the target genes of trans-acting lncRNAs. The KEGG enrichment analysis revealed that the lncRNA target genes were significantly enriched in 44 pathways (*p* < 0.05). These pathways included chemical carcinogenesis—DNA adducts (ko05204), complement and coagulation cascades (ko04610), steroid hormone biosynthesis (ko00140), PPAR signaling pathway (ko03320), glycolysis/gluconeogenesis (ko00010), fat digestion and absorption (ko04975), and glucagon signaling pathway (ko04922) (Figure 3A).

A total of 704 GO terms were identified as significantly enriched (*p* < 0.05) in the GO enrichment analysis. Among these terms, 42 were associated with cellular components, 108 were associated with molecular functions, and 554 were associated with biological processes. Among the significantly enriched GO terms, the highest number is observed in biological processes, including some important GO terms related to high-altitude hypoxia adaptation, such as response to external stimulus (GO:0009605), oxoacid metabolic process (GO:0043436), steroid metabolic process (GO:0008202), lipid transport (GO:0006869), acute inflammatory response (GO:0002526), oxidation-reduction process (GO:0055114), and angiogenesis (GO:0001525) (Figure 3B). Additionally, we have identified some target genes of DE lncRNAs that overlap with DEPs. These target genes may play a crucial role in the biological process of altitude hypoxia adaptation in Tibetan sheep (Appendix A).

### 2.3. Functional Assessment of DEPs in Tibetan Sheep at Different Altitudes

To investigate the role of DEPs in the adaptation of Tibetan sheep cardiac tissues to plateau hypoxia, we conducted GO and KEGG analyses on DEPs from three comparison groups at varying altitudes. The results revealed significant enrichment of the top 20 GO terms in all three comparative groups (Figure 4A,C,E). These functional enrichment findings were consistent with those of differentially expressed long non-coding RNA (lncRNA) target genes. Notably, the significantly enriched DEPs included numerous GO terms associated with the adaptation of Tibetan sheep to high-altitude hypoxia.

In the TS25 and TS35 groups, the significantly enriched GO terms were humoral immune response (GO:0006959), protein activation cascade (GO:0072376), pH elevation (GO:0045852), response to steroid hormone (GO:0048545), response to oxygen-containing compound (GO:1901700), reactive oxygen species metabolic process (GO:0072593), and regulation of vasodilation (GO:0042312). Similarly, the TS35 and TS45 groups exhibited significant enrichment in reactive oxygen species metabolic processes (GO:0072593), glycosaminoglycan biosynthetic processes (GO:0006024), peroxidase activity (GO:0004601), inflammatory response (GO:0006954), response to steroid hormone (GO:0048545), vasodilation (GO:0042311), and hemopoiesis (GO:0030097). Lastly, the TS25 and TS45 groups showed significant enrichment in regulation of cell proliferation (GO:0042127), regulation of lymphocyte proliferation (GO:0050670), regulation of leukocyte proliferation (GO:0070663), humoral immune response (GO:0006959), immune effector process (GO:0002252), reactive oxygen species metabolic process (GO:0072593), and response to lipid (GO:0033993).

Through KEGG enrichment analysis, we also identified several key pathways that are associated with hypoxic adaptation in Tibetan sheep. These pathways include cholesterol metabolism (ko04979), metabolic pathways (ko01100), the PPAR signaling pathway (ko03320), steroid hormone biosynthesis (ko00140), and hematopoietic cell lineage (ko04640), which were found to be significantly enriched in the TS25 and TS35 groups (Figure 4B). Additionally, allograft rejection (ko05330), asthma (ko05310), viral myocarditis (ko05416), the PPAR signaling pathway (ko03320), and cholesterol metabolism (ko04979) were significantly enriched in the TS25 and TS45 groups (Figure 4D). Furthermore, microRNAs in cancer (ko05206), tryptophan metabolism (ko00380), hematopoietic cell lineage (ko04640), peroxisome (ko04146), and metabolic pathways (ko01100) were significantly enriched in the TS35 and TS45 groups (Figure 4F).

### 2.4. Trend Analysis of DE LncRNAs and DEPs

To investigate the expression patterns of DE lncRNAs and DEPs in Tibetan sheep at three consecutive altitudes, we conducted trend clustering analyses on all DE lncRNAs and DEPs. Meanwhile, the red/gray shading indicates the significantly enriched profiles of DE lncRNAs or DEPs identified in the trend analysis (Figure 5). Based on their distinct expression trends, the DE lncRNAs were categorized into eight profiles, two of which exhibited significant enrichment (*p* < 0.05). These profiles included down-regulated (profile 3) and up-regulated (profile 7), comprising 44 and 23 DE lncRNAs, respectively (Figure 5A, Appendix A). Furthermore, profile 5 and profile 3 were significantly enriched (*p* < 0.05) in the eight module species formed by DEPs, encompassing 88 and 63 DEPs, respectively (Figure 5B, Appendix A). Notably, among these significant enrichment patterns, DEPs in profile 3 displayed an overall decreasing trend with increasing elevation, while DEPs in profile 5 exhibited the highest expression in the TS35 group.

### 2.5. Protein–Protein Interaction Network Analysis of DEPs

To characterize the modifications in protein interaction networks in Tibetan sheep heart tissue at elevation, we performed a PPI analysis of DEPs in three exceptional evaluation companies at elevation using the online tool STRING. In the TS25 and TS35 comparison groups, 87 constituted an interaction network related to the steroid metabolic processes (ACAA1, APOA4, APOA1, HSL, APOA2, PGFS, and HSD11B1), cholesterol metabolism (APOC3, APOE, APOA2, and APOH), and PPAR signaling pathways (APOC3, FABP4, APOA2, and EHHADH) (Figure 6A). Moreover, 36 constituted an interaction network related to the oxidation-reduction process (ADH1C, ALDH1A1, MAOB, GPX3, and AOC1) and inflammatory response (RIPK1, IDO1, CRP, and ORM1) between the TS25 and TS45 groups (Figure 6B). Moreover, 106 constituted an interaction network related to the complement and coagulation cascades (SERPINB2, CFD, F9, MBL2, and CLU) and the oxidation-reduction process (for instance, EHHADH, MAOB, CAT, and GPX3) between the TS35 and TS45 groups (Figure 6C).

### 2.6. Correlation between Transcriptome and Proteome

To search for genes/proteins co-existing in transcriptome and proteome data, we carried out a comparative evaluation of DEPs, DEGs, and DE lncRNA target genes. The results confirmed that 41 genes/proteins coexisted in three clusters (Figure 7A), and some key pathways and GO terms had been determined by constructing the interaction networks of these shared genes/proteins, such as the PPAR signaling pathway (APOC3, FABP4, and APOA2), glycolysis/gluconeogenesis (ALDOB, FBP1, and ADH1C), cholesterol metabolism (APOC3, APOA2, and APOH), and regulation of lipase activity (APOA4, APOC3, APOA2, and APOH) (Figure 7B,C). Moreover, a lncRNA-mRNA co-expression network was constructed to investigate the targeting of DE lncRNAs to these shared genes/proteins (Figure 7D). It was shown that the regulation of mRNA by lncRNAs may be involved in Tibetan sheep’s adaptation to hypoxic plateau environments. For instance, lncRNA ENSOART00020004609, ENSOART00020025894, and ENSOART00020036371 may target and regulate 26, 28, and 27 shared genes/proteins, respectively.

### 2.7. LncRNA Expression Validated by RT-qPCR

To explore the accuracy of the sequencing results, 10 lncRNAs were randomly selected, and their expression was detected using RT-qPCR assays. The results indicated that all experimental results were consistent with the sequencing data (Figure 8).

### 2.8. Correlation Analysis between Genes and Blood Physiological and Biochemical Indicators

To investigate the role of DE lncRNAs in the hypoxic adaptation mechanism of Tibetan sheep living at high altitudes, we carefully selected specific genes (APOA4, APOC3, APOA2, APOH, ALDOB, FBP1, ADH1C, and FABP4) that are closely associated with these adaptive processes. Subsequently, we generated a Sankey diagram to visually represent the significantly enriched KEGG pathways or GO terms related to these genes (Figure 9A). Additionally, we constructed a correlation heatmap using Pearson correlations to identify potential relationships between the genes and blood physiological and biochemical indexes (Figure 9B). Notably, FABP4 showed a positive correlation with the concentration of bicarbonate (HCO_3_^−^). Lactate dehydrogenase (LDH) and lactate dehydrogenase isoenzymes (LDH1) exhibited positive correlations with multiple genes, excluding FABP4. Furthermore, we observed negative correlations between creatine kinase (CK) and superoxide dismutase (SOD) with various genes and lncRNAs.

## 3. Discussion

Low oxygen is a prominent characteristic of a plateau climate, and it significantly impacts human health as well as animal production and reproduction performance. Tibetan sheep, a native breed residing in the Qinghai-Tibetan Plateau for an extended period, offers valuable insights into the adaptive mechanisms of the low-oxygen environment. Studying these mechanisms can help uncover crucial information regarding low oxygen adaptation, which can aid in predicting, preventing, and treating hypoxic diseases in both humans and animals on the plateau. Research has shown that when exposed to low-oxygen environments, animals could enhance the oxygen-carrying capacity of their blood vessels and blood, thereby increasing the oxygen content in their tissues [32,33]. Simultaneously, their blood physiology, biochemistry, and other indicators undergo corresponding changes. In this study, as altitude increased, PO_2_ and SO_2_ exhibited a declining pattern, while HGB and HCT increased, indicating that Tibetan sheep enhance their blood’s oxygen-carrying capacity by increasing the number of red blood cells and hemoglobin. This adaptation enables them to thrive in hypoxic environments. Moreover, Tibetan sheep living at 4500 m above sea level displayed significantly higher levels of CK, CK-MB (creatine kinase isoenzymes), LDH, and LDH1 (lactate dehydrogenase isoenzymes) compared to those at lower altitudes. CK and CK-MB are crucial energy-regulating enzymes in the myocardium [34], suggesting that Tibetan sheep at higher altitudes may possess a greater ability to produce ATP, facilitating better adaptation to low-oxygen conditions. Furthermore, with the rise in altitude, there is a decrease in the oxygen levels present in the atmosphere. This decrease in oxygen concentration prompts an increase in the proportion of glycolysis, ultimately leading to the generation of significant quantities of lactic acid. Simultaneously, the elevation in LDH and LDH1 levels in the bloodstream of Tibetan sheep at higher altitudes enables the reversible catalysis of lactic acid into pyruvic acid. This enzymatic process serves to mitigate the detrimental impact of lactic acid on the animal’s physiological functions, including those of the heart and liver [35].

Numerous successful scientific studies have confirmed the maturity of transcriptome sequencing and proteomics technology, which has gradually been applied to the investigation of animal adaptation to high-altitude hypoxia [36,37]. However, there is still a lack of research on the integration of transcriptome and proteomics data to analyze the hypoxic adaptation of sheep. In this study, we have identified a total of 248 DE lncRNAs and 279 DE lncRNA target genes in the heart tissues of Tibetan sheep at three different altitudes. By conducting enrichment analysis on the target genes, we have also discovered pathways and terms associated with high-altitude hypoxic adaptation, such as the PPAR signaling pathway, glycolysis/glycogenesis, the oxoacid metabolic process, and angiogenesis. In hypoxic environments, the decrease in oxygen concentration leads to an increase in the proportion of ATP produced by animal bodies through glycolysis [38]. Research has revealed that the PPAR signaling pathway is closely associated with energy production processes, including lipid metabolism and glucose metabolism [39], indicating that the PPAR signaling pathway also plays a crucial role in the adaptation of Tibetan sheep to high-altitude hypoxia. Furthermore, our analysis of the cardiac proteome of Tibetan sheep showed a significant enrichment of reactive oxygen species metabolic processes in the three comparative groups at different altitudes. This highlights the importance of this process in hypoxic adaptation. By integrating transcriptome and proteome data, we have identified 41 key genes/proteins that are present in all three datasets: DE LncRNAs target genes, DEPs, and DEGs (Figure 7A). Through the exploration of the functions of these shared genes/proteins, we have identified pathways that align with previous transcriptome and proteome findings, such as the PPAR signaling pathway and glycolysis/gluconeogenesis, among others (Figure 7C).

The construction of PPI network diagrams is an effective method for identifying key proteins that play crucial roles in protein functions. In this study, we utilized PPI network analysis to identify significant proteins such as ADH1C, APOH, FABP4, APOA4, APOC3, APOA2, and ALDOB, among others. The *ADH1C* gene is closely associated with lipid metabolism and deposition [40]. Previous research has demonstrated that an increase in ADH1C protein expression leads to elevated intramuscular fat (IMF) content in beef [41]. Our study revealed that the *ADH1C* gene is significantly enriched in metabolic pathways, including glycolysis/gluconeogenesis and carboxylic acid metabolic processes. In hypoxic conditions, the proportion of glycolysis increases, and the resulting lactic acid can be converted into pyruvate by lactate dehydrogenase. Pyruvate is further converted into acetyl CoA, which enters the tricarboxylic acid cycle [42] to generate ATP, providing energy for Tibetan sheep. Furthermore, we found that the polymorphism of the Apolipoprotein H (*APOH*) gene is associated with elevated levels of lipoprotein (a), an independent risk factor for cardiovascular disease [43]. The incidence of ischemic cardiovascular and cerebrovascular diseases is significantly higher in high-altitude and hypoxic areas compared to plain areas [44,45,46,47]. Fatty acid-binding protein 4 (FABP4) is a lipid-binding protein that belongs to the intracellular family. It is primarily expressed in adipose tissue and is also found in the heart [48]. FABP4 has been identified as a potential biomarker for atherosclerosis [49] and is strongly associated with inflammation, obesity, diabetes, and cardiovascular disease [50,51]. Research has demonstrated that apolipoprotein A4 (APOA4), apolipoprotein C3 (APOC3), and apolipoprotein A2 (APOA2) are apolipoproteins that play crucial roles in lipid metabolism. They are primarily involved in encoding the plasma lipid transport system and have been significantly linked to lipoprotein and cardiovascular diseases [52,53]. Additionally, these three proteins are closely associated with HDL cholesterol (HDL-C) metabolism [54]. HDL-C is believed to have a protective effect against atherosclerosis and aid in the transportation of cholesterol from peripheral tissues to the liver for further processing [55,56].

Additionally, the upregulation of aldolase B (ALDOB) induced by fructose aids in the increased production of methylglyoxal (MG) in blood vessels, resulting in vascular remodeling [57]. Vascular remodeling refers to the adaptive changes in the structure of blood vessels under the influence of hemodynamics. However, excessive vascular remodeling can lead to arterial wall hardening, causing the loss of its ability to properly adjust its size and ultimately leading to atherosclerosis in both humans and animals [58,59]. The proteome analysis results revealed that the expression level of ALDOB protein in the heart tissue of Tibetan sheep in the TS35 group (mean = 314.1354656) was significantly higher than that of Tibetan sheep in the TS25 group (mean = 49.59152269) (*p* < 0.05). Conversely, the expression level of ALDOB protein in the heart tissue of Tibetan sheep in the TS45 group (mean = 42.02183676) was significantly lower than that of the TS35 group (*p* < 0.05). This indicates that Tibetan sheep at an altitude of 4500 m may have undergone long-term hypoxic adaptation, resulting in a significant reduction in the adverse effects caused by vascular remodeling. Consequently, the protein expression level of ALDOB in their hearts is comparable to that of Tibetan sheep at an altitude of 2500 m.

By employing trend analysis methods, we can investigate the expression patterns of DE lncRNAs and DEPs in the heart tissue of Tibetan sheep at three altitudes. This enables us to identify the genetic information that plays a crucial role in the hypoxic adaptation process of Tibetan sheep. Consequently, we utilized the key DE lncRNAs and shared DE genes/proteins identified from various modules of trend analysis to construct an lncRNA-mRNA co-expression network. This network allows us to explore the relationship between lncRNAs and functional genes/proteins. The findings suggest a strong correlation between DE lncRNAs and DE genes/proteins, indicating that lncRNAs also have a significant involvement in the hypoxic process of Tibetan sheep at high altitudes.

The correlation analysis between genes and blood physiological and biochemical indicators reveals a significant correlation among these indicators, suggesting a reciprocal influence. This finding enhances our comprehension of the adaptation mechanism of Tibetan sheep to high-altitude hypoxia. For instance, LDH and LDH1 were associated with multiple genes/proteins, including APOH, APOA4, APOC3, APOA2, and partial lncRNAs. LDH is a crucial enzyme involved in the anaerobic glycolysis process [60], while apolipoprotein is a key component of plasma lipoprotein that plays a role in various processes such as inflammation inhibition, oxidative stress prevention, tissue remodeling, lipid metabolism, energy consumption, and more [61,62]. As altitude increased, the environment’s PO_2_ decreased, leading to an increased proportion of glycolysis in animal organisms and a higher likelihood of lipid metabolism and inflammation. Therefore, it is reasonable to expect a positive correlation between apolipoprotein and LDH in this study. However, further investigation is needed to understand the specific mechanism of interaction between these two factors.

## 4. Materials and Methods

### 4.1. Ethics Statement

This study protocol was approved by the Ethics Committee of Gansu Agricultural University (protocol code GSAU-Eth-AST-2021-001, approval date: 15 January 2021).

### 4.2. Animals and Sample Collection

The same heart tissue samples were used, as described by Wen et al. (2022) [63]. In this study, 4 healthy 3.5-year-old Tibetan sheep ewes were selected at an altitude of 2500 m (Zhuoni, Gansu, China) (TS25), 3500 m (Haiyan, Qinghai, China) (TS35), and 4500 m (Zhiduo, Yushu, Qinghai, China) (TS45), respectively, according to their distribution. The sheep were fasted for 12 h before slaughter, and heart tissue (left ventricle) was collected in cryogenic vial tubes after slaughter, immediately transferred to liquid nitrogen, and stored at −80 °C for subsequent transcriptomics and proteomic sequencing. Animals from each group were slaughtered at their respective altitudes. All ewes were slaughtered in a humane manner, adhering to Islamic practices, which involved exsanguination, peeling, and splitting down the midline following standard operating procedures. The experiment was approved by the Animal Care Committee at Gansu Agricultural University, with due consideration given to animal welfare and conditions during the use of experimental animals. The blood physiological and biochemical indexes of these sheep were investigated in the study described in Table 2 [63].

### 4.3. Total RNA Extraction, cDNA Library Construction, and Sequencing

Total RNA was isolated using the Trizol kit (Invitrogen, Carlsbad, CA, USA). The quality of the RNA was assessed using an Agilent 2100 Bioanalyzer (Agilent Technologies, Palo Alto, CA, USA) and confirmed with RNase-free agarose gel electrophoresis. After RNA extraction, ribosomal RNAs (rRNAs) were eliminated, while messenger RNAs (mRNAs) and non-coding RNAs (ncRNAs) were retained. The enriched mRNAs and ncRNAs were fragmented using a fragmentation buffer and then reverse transcribed into complementary DNA (cDNA) using random primers. Second-strand cDNA was synthesized with DNA polymerase I, RNase H, dNTP (dUTP was used instead of dTTP), and buffer. The resulting cDNA fragments were purified using the QiaQuick PCR extraction kit (Qiagen, Venlo, The Netherlands), end-repaired, and poly(A) tails were added. Subsequently, they were ligated to Illumina sequencing adapters. Uracil-N-glycosylase (UNG) was employed to digest the second-strand cDNA. The digested product was size-selected via agarose gel electrophoresis, PCR amplified, and sequenced by Gene Denovo Biotechnology (Guangzhou, China) using the Illumina HiSeqTM 4000 platform (San Diego, CA, USA).

### 4.4. Quality Control, Reference Genome Alignment, and Differential Expression Analysis

To obtain high-quality clean reads, the reads were further filtered using fastp [64] (version 0.18.0). The filtering parameters included the removal of reads containing adapters, reads containing more than 10% of unknown nucleotides (N), and low-quality reads containing more than 50% of low-quality bases (Q-value ≤ 20). The filtered clean reads were then aligned to the sheep reference genome (ncbi_GCA_002742125.1_Oar_v1.0) using HISAT2 (v2.1.0). Gene expression levels were quantified using FPKM (fragments per kilobase of transcript sequence per million base pairs sequenced).

Transcripts were assembled using the Stringtie (v1.3.4) [65] software, combining the results from the HISAT2 [66] software comparisons. Transcripts with uncertain strand orientation were removed, and the remaining transcripts, with a retained length of ≥200 bp and a number of exons ≥ 2, were used for lncRNA screening. The coding ability of new transcripts was predicted using CPC2 (v0.1) [67] and CNCI (v2) [68] software. The intersection of these transcripts without coding potential was considered lncRNA. Differential expression (DE) profiles of lncRNAs and mRNAs in the heart tissue of Tibetan sheeps were analyzed using DESeq2 (v1.20.0) [69] software (FDR < 0.05 and |log_2_FC| > 1). In addition, DE lncRNAs were predicted to target genes using cis and trans methods. Cis: genes with a distance of 10 kb upstream or downstream from lncRNA. Trans: expression correlation (Pearson) between lncRNAs and protein-coding genes was analyzed to predict the target genes of trans.

### 4.5. Sample Preparation for Mass Spectrometry

Sample pre-treatment involves various processes such as protein extraction, denaturation, reductive alkylation, enzymatic digestion, and desalting of peptides. In this study, tissue samples were pre-treated using the iST sample pre-treatment kit (PreOmics, Planegg, Germany). The samples were ground in liquid nitrogen, followed by the addition of 50 µL of lysis solution. Subsequently, the samples were heated at 95 °C and 1000 rpm for 10 min. After cooling to room temperature, trypsin digestion buffer was added and incubated at 37 °C and 500 rpm for 2 h. The enzymatic reaction was stopped by adding the termination buffer. The peptides were desalted using the iST cartridge provided in the kit and eluted with elution buffer (2 × 100 µL). The eluted peptides were then vacuum-dried and stored at −80 °C. The samples were brought to room temperature and incubated with trypsin digestion buffer at 37 °C and 500 rpm for 2 h. The reaction was stopped using the termination buffer.

The peptide mixture was dissolved again in buffer A (buffer A: 20 mM ammonium formate in water at pH 10.0, adjusted with ammonium hydroxide). The mixture was then separated by high pH using the Ultimate 3000 system (Thermo Fisher scientific, Waltham, MA, USA) connected to a reverse phase column (XBridge C18 column, 4.6 mm × 250 mm, 5 μm, Waters Corporation, Milford, MA, USA). The high pH separation process involved a linear gradient, starting from 5% B and increasing to 45% B over a period of 40 min (buffer B: 20 mM ammonium formate in 80% ACN at pH 10.0, adjusted with ammonium hydroxide). The column was re-equilibrated for 15 min at the initial condition. The flow rate of the column was maintained at 1 mL/min, and the temperature was kept at 30 °C. A total of ten fractions were collected, and each fraction was dried in a vacuum concentrator for the subsequent step.

### 4.6. Construction and Spectral Library of Qualitative Databases in Data-Dependent Acquisition

The peptides were dissolved again in 30 μL of solvent A (solvent A: 0.1% formic acid in water). The analysis was performed using an Orbitrap Fusion Lumos coupled to an EASY-nLC 1200 system (Thermo Fisher Scientific, Waltham, MA, USA) with on-line nanospray LC-MS/MS. A 3 μL sample of the peptide was loaded onto the analytical column (Acclaim PepMap C18, 75 μm × 25 cm) and separated using a 120-min gradient, ranging from 5% to 35% solvent B (0.1% formic acid in CAN). The flow rate of the column was maintained at 200 nL/min, and the column temperature was set at 40 °C. An electrospray voltage of 2 kV was applied at the inlet of the mass spectrometer. The mass spectrometer operated in data-dependent acquisition mode, automatically switching between MS and MS/MS modes. The following parameters were used: (1) MS: scan range (*m*/*z*) = 350–1200; resolution = 120,000; AGC target = 400,000; maximum injection time = 50 ms; Filter Dynamic Exclusion: exclusion duration = 30 s; (2) HCD-MS/MS: resolution = 15,000; AGC target = 50,000; maximum injection time = 35 ms; collision energy = 32.

The raw data from DDA (data-dependent acquisition) were processed and analyzed using Spectronaut version 14.0 (Biognosys AG, Zurich, Switzerland) with the default settings to generate an initial target list. Spectronaut was configured to search the database of the ovine genome assembly (ncbi_GCA_002742125.1_Oar_v1.0) as well as a contaminant database, assuming trypsin as the digestion enzyme. Carbamidomethyl (C) was designated as the fixed modification, while oxidation (M) was designated as the variable modification. A Q value (FDR) [70] cutoff of 1% was applied at the precursor and protein levels.

### 4.7. DIA Data Collection and Differential Expression Analysis

The mass spectrometer operated in data-independent acquisition mode, automatically switching between MS and MS/MS modes. The parameters used were as follows: (1) MS: scan range (*m*/*z*) = 350–1200; resolution = 120,000; AGC target = 1 × 10^6^; maximum injection time = 50 ms; (2) HCD-MS/MS: resolution = 30,000; AGC target = 1 × 10^6^; collision energy = 32; stepped CE = 5%. (3) DIA was performed with a variable isolation window, with each window overlapping by 1 *m*/*z*, and the window number was 60.

The default settings of Spectronaut X (Biognosys AG, Zurich, Switzerland) were used to analyze the raw data of DIA. The q-value (FDR) cutoff for both precursor and protein levels was set at 1%. Decoy generation was configured as mutated. All precursors that passed the filters were selected for quantification, and the major group quantities were calculated using the average of the top 3 filtered peptides. We identified differentially expressed proteins (DEPs) by applying a q-value < 0.05 and fold-change >1.5 (or <0.58) criterion to compare the Tibetan sheep at varying altitudes.

### 4.8. Functional Enrichment Analysis

Gene ontology (http://geneontology.org/ (accessed on 20 February 2023)) (GO) and Kyoto Encyclopedia of Genes and Genomes (https://www.genome.jp/kegg/ (accessed on 20 February 2023)) (KEGG) were used to explore the function of DEPs and target genes of DE lncRNAs. In addition, protein–protein interaction (PPI) analyses of DEPs were implemented using the STRING (http://string.embl.de/ (accessed on 23 February 2023)), where the confidence score was set to high (score > 0.7). To investigate the expression patterns of DEPs and target genes of DE lncRNAs, we used Short Time Series Expression Miner (STEM) software (Version 1.3.13) for cluster analysis [71]. Furthermore, a lncRNA-mRNA co-expression network was constructed using Cytoscape 3.7.1 [72].

### 4.9. RT-qPCR Analysis

To verify the accuracy of the sequencing results, we randomly selected 10 DE lncRNAs for validation analysis using RT-qPCR. PCR primers were designed using primer 5.0 and then checked for primer specificity by the NCBI primer blast (Table 3). The relative expression levels of lncRNAs were calculated by the 2^−(∆∆Ct)^ method.

### 4.10. Statistical Analyses

Statistical analysis was performed using SPSS 22.0 with one-way analysis of variance (ANOVA). The results were presented as mean ± SD, and *p* < 0.05 was considered statistically significant. GraphPad Prism 8.0.1 was used for graphing. Pearson correlation analysis was used to judge the correlation between genes and blood physiological and biochemical indicators.

## 5. Conclusions

In summary, our study integrated transcriptomic and proteomic data from the hearts of Tibetan sheep at different altitudes, revealing the crucial role of the PPAR signaling pathway, glycolysis/glycogenesis, oxoacid metabolic process, and angiogenesis in their adaptation to hypoxic environments. Through transcriptomics and proteomics analysis, we identified numerous DEGs and DEPs. Notably, *APOA2*, *ALDOB*, *FBP1*, *FABP4*, and *ADH1C* genes were found to be associated with oxidative metabolic processes, while *APOH* was associated with angiogenesis. By integrating transcriptome and proteomic data, we discovered a total of 41 shared DEGs/DEPs among the dataset of DE lncRNA target genes, DEGs, and DEPs. Among these, *APOC3*, *FABP4*, and *APOA2* genes were linked to fat synthesis, whereas ALDOB, FBP1, and ADH1C were associated with glycolysis/glycogenesis. The identification of these target genes and shared genes/proteins suggested that lncRNAs and genes likely play a pivotal role in the adaptation process of Tibetan sheep to high-altitude hypoxia.

## Figures and Tables

**Figure 1 ijms-25-00385-f001:**
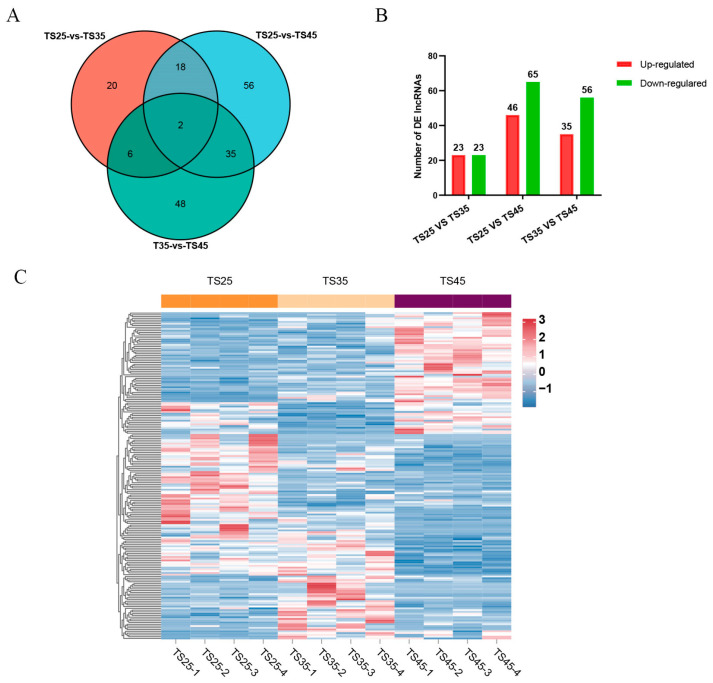
Analysis of differentially expressed (DE) lncRNAs in the hearts of Tibetan sheep at varying altitudes. (**A**) Venn diagram of DE lncRNAs in the three comparison groups; (**B**) histogram of DE lncRNAs in the three comparative groups; (**C**) clustering heatmap of DE lncRNAs in the three consecutive altitudes.

**Figure 2 ijms-25-00385-f002:**
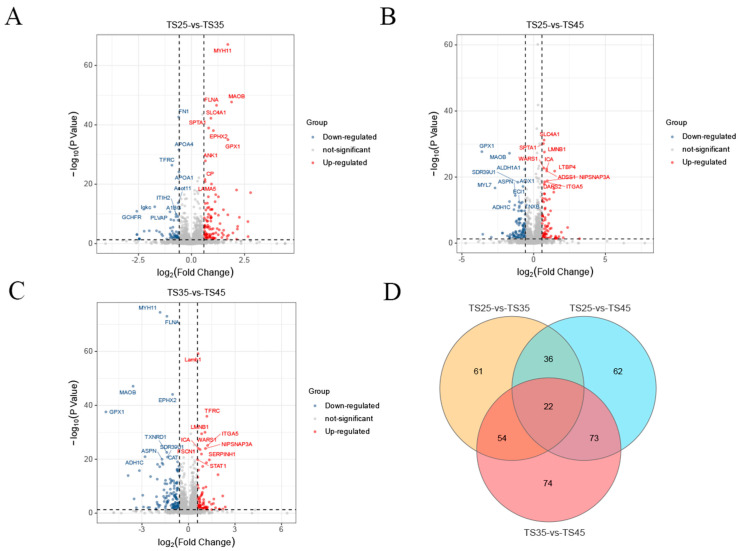
Analysis of differentially expressed (DE) proteins in the hearts of Tibetan sheep at varying altitudes. (**A**) Volcano map of DEPs in the T25 and T35 comparison groups; (**B**) Volcano map of DEPs in the T25 and T45 comparison groups; (**C**) Volcano map of DEPs in the T25 and T45 comparison groups; (**D**) Venn diagram of DE lncRNAs in the three comparison groups.

**Figure 3 ijms-25-00385-f003:**
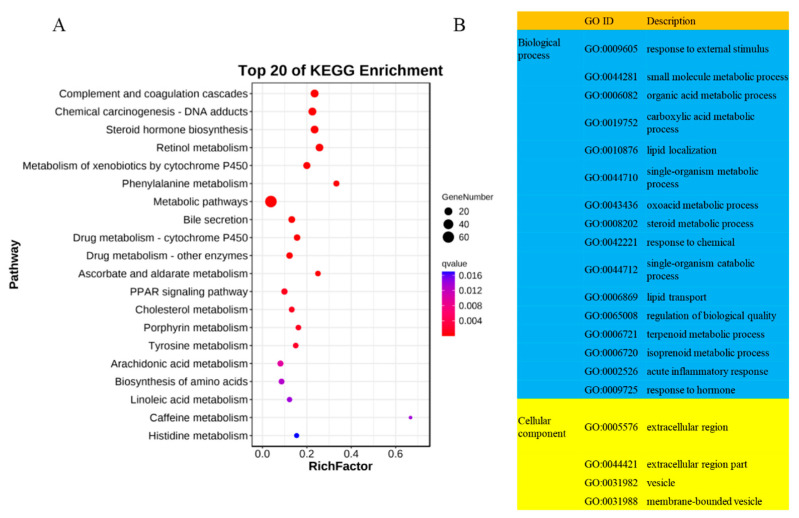
Functional enrichment analysis of DE lncRNA target genes in Tibetan sheep at different altitudes. (**A**) Bubble chart of the Kyoto encyclopedia of genes and genomes (KEGG) enrichment analysis; (**B**) gene ontology (GO) enrichment analysis of lncRNA target genes.

**Figure 4 ijms-25-00385-f004:**
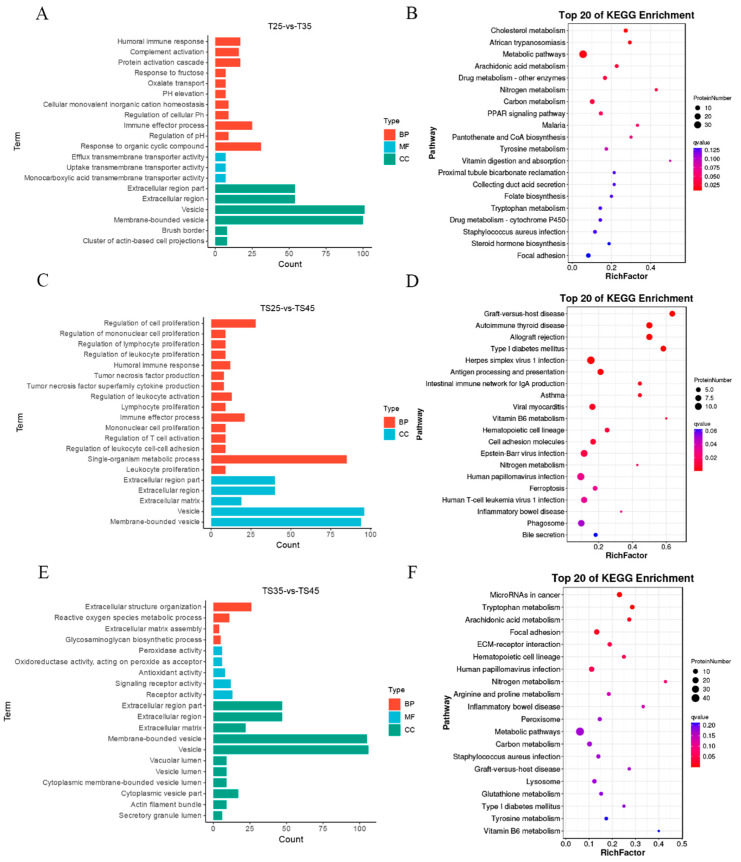
Functional enrichment analysis of DEPs in Tibetan sheep at different altitudes. (**A**,**C**,**E**) GO enrichment analysis of DEPs in TS25 and TS35, TS25 and TS45, and TS35 and TS45 comparison group; (**B**,**D**,**F**) KEGG enrichment analysis of DEPs in TS25 and TS35, TS25 and TS45, and TS35 and TS45 comparison groups.

**Figure 5 ijms-25-00385-f005:**
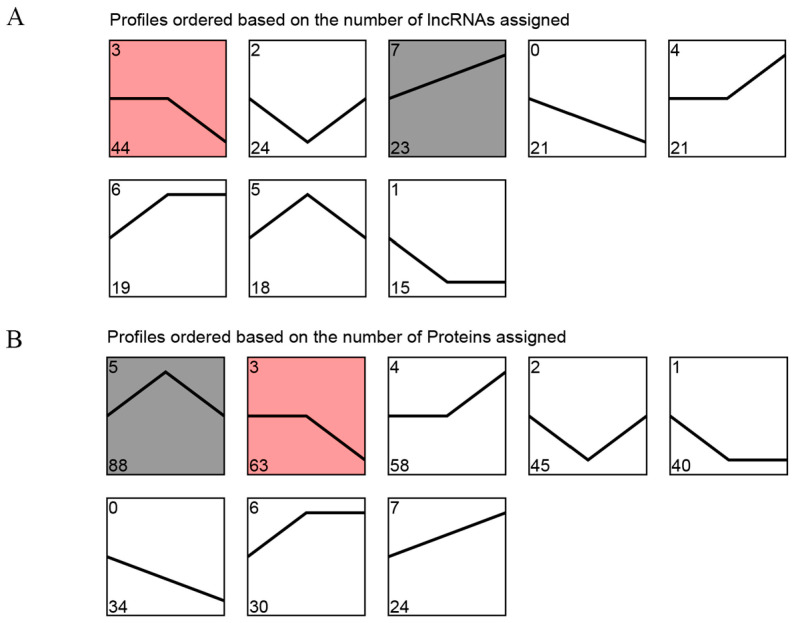
Trend analysis of DE lncRNAs and DEPs in Tibetan sheep at different altitudes. (**A**) Trend analysis of DE lncRNAs; (**B**) trend analysis of DEPs. The red/gray shading indicates the significantly enriched profiles of DE lncRNAs or DEPs identified in the trend analysis.

**Figure 6 ijms-25-00385-f006:**
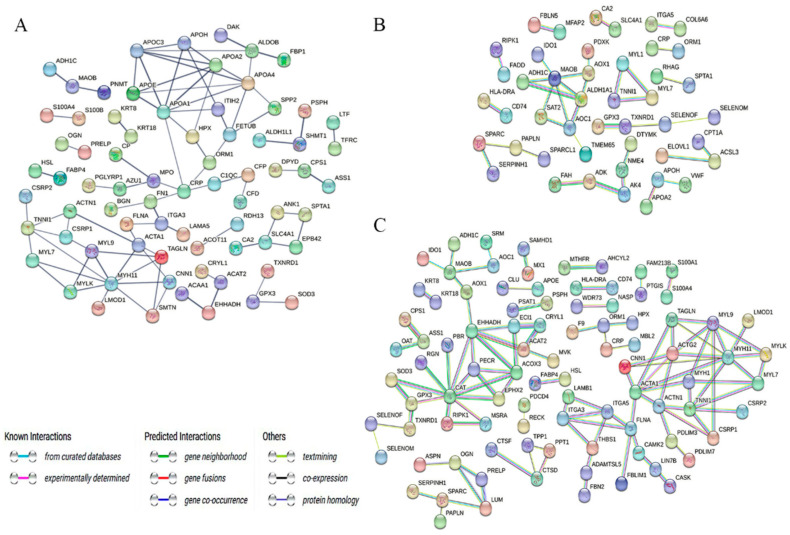
Protein–protein interaction network analysis of DEPs. (**A**) Protein–protein interaction network analysis in the TS25 and TS35 comparison groups; (**B**) protein–protein interaction network analysis in the TS25 and TS45 comparison groups; (**C**) protein–protein interaction network analysis in the TS35 and TS45 comparison groups.

**Figure 7 ijms-25-00385-f007:**
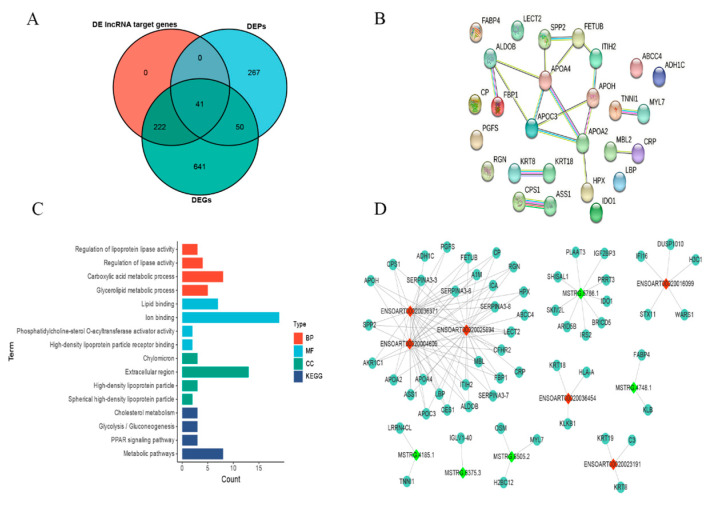
Combined analysis of the transcriptome and proteome. (**A**) Venn diagram of DE lncRNA target genes, DEPs, and DEGs; (**B**) protein–protein interaction networks of shared genes/proteins in transcriptome and proteome data; (**C**) functional enrichment analysis of shared genes/proteins in transcriptome and proteome data; (**D**) co-expression network of lncRNA-mRNA. Red triangles represent up-regulated lncRNAs, while light green triangles represent down-regulated lncRNAs. Dark green circles indicate shared genes/proteins targeted by both up-regulated and down-regulated lncRNAs.

**Figure 8 ijms-25-00385-f008:**
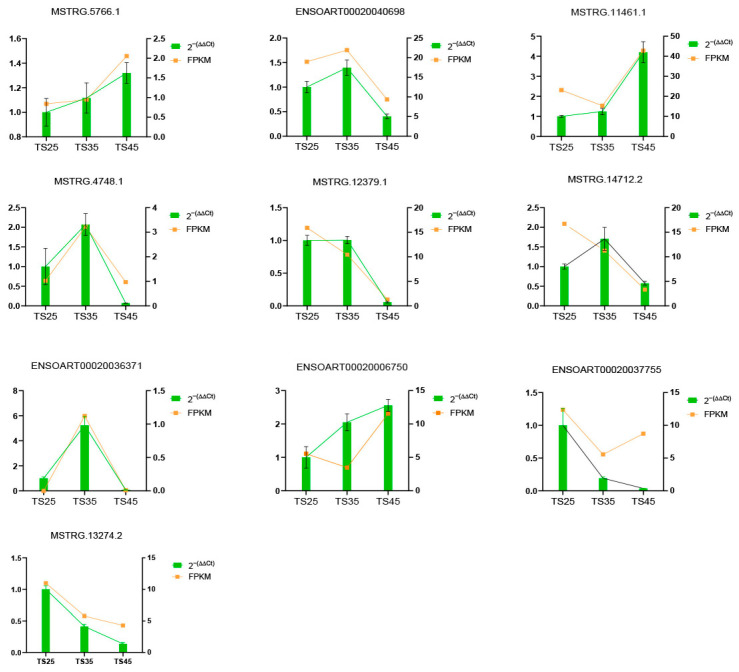
Comparison of lncRNA expression levels between RNA-Seq and RT-qPCR. RT-qPCR data were presented as means ± S.D., with 2^−(∆∆Ct)^ representing the RT-qPCR result. FPKM was used to represent the RNA-Seq result.

**Figure 9 ijms-25-00385-f009:**
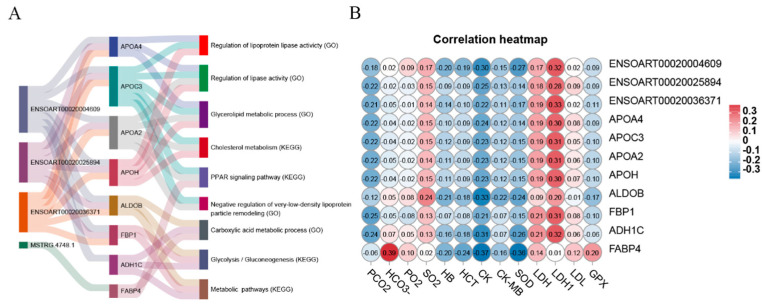
Correlation analysis. (**A**) Sankey diagram of KEGG pathways and GO terms for DE lncRNA target genes of adaptability to Highland Hypoxia; (**B**) Pearson correlation analysis between genes and blood physiological and biochemical indexes.

**Table 1 ijms-25-00385-t001:** Clean reads and reference genome alignment results.

Sample	Average Raw Reads	Average Clean Reads	Average Remaining Clean Reads	Average Mapped Reads (%)
TS25	73,744,200	73,433,514 (99.58%)	73,324,286	63,119,866 (86.08%)
TS35	72,917,734	72,677,193 (99.67%)	72,526,068	62,588,618 (86.30%)
TS45	89,686,072	89,332,025 (99.61%)	89,184,858	78,171,553 (87.65%)

**Table 2 ijms-25-00385-t002:** Blood physiological and biochemical indexes of Tibetan sheep at different altitudes.

**Blood Physiological Indexes**	**Tibetan Sheep**		
	**TS25**	**TS35**	**TS45**
Oxygen pressure, PO_2_ (mmHg)	37.75 ± 2.02 a	34.00 ± 1.78 a	28.50 ± 1.04 b
Oxygen saturation, SO_2_ (%)	72.00 ± 0.82 a	65.75 ± 0.91 b	59.75 ± 0.83 c
Hemoglobin, HGB (g/dL)	11.60 ± 0.44 c	13.05 ± 0.44 b	15.80 ± 0.37 a
Hematocrit, HCT (%PCV)	34.75 ± 0.85 c	39.50 ± 0.87 b	50.50 ± 0.87 a
Potential of hydrogen, pH	7.37 ± 0.03 a	7.33 ± 0.02 a	7.31 ± 0.01 a
Carbon dioxide pressure, PCO_2_ (mmHg)	55.70 ± 1.42 a	40.75 ± 1.19 b	34.92 ± 1.64 c
Concentration of bicarbonate, HCO_3_^−^ (mmol/L)	27.35 ± 0.79 a	25.38 ± 0.83 a	25.03 ± 0.81 a
Base excess, BE (mmol/L)	3.00 ± 0.41 a	3.25 ± 0.48 a	3.75 ± 0.48 a
**Blood Biochemical Indexes**	**TS25**	**TS35**	**TS45**
Creatine kinase, CK (U/L)	256.60 ± 4.43 c	300.34 ± 4.31 b	499.62 ± 6.38 a
Creatine kinase isoenzymes, CK-MB (U/L)	42.12 ± 1.86 c	49.70 ± 1.90 b	58.91 ± 2.12 a
Lactate dehydrogenase, LDH (U/L)	616.57 ± 7.68 c	833.31 ± 6.37 b	906.08 ± 5.09 a
Lactate dehydrogenase isoenzymes, LDH1 (U/L)	131.34 ± 4.25 b	156.68 ± 4.48 a	164.78 ± 5.23 a
Superoxide dismutase, SOD (U/mL)	198.03 ± 4.77 b	206.28 ± 4.59 b	244.20 ± 4.65 a
Glutathione peroxidase, GPX (U/mL)	54.76 ± 3.03 b	64.83 ± 2.46 ab	69.06 ± 5.49 a
Low-density lipoprotein, LDL (mmol/L)	1.26 ± 0.11 a	1.23 ± 0.14 a	1.23 ± 0.12 a

The data shown in the table are means ± SEM, and different lowercase letters indicate that the difference was significant.

**Table 3 ijms-25-00385-t003:** Primers of RT-qPCR.

LncRNAs	Forward (5′ → 3′)	Reverse (5′ → 3′)
MSTRG.5766.1	TGAAGCCAGGTCCCTCCTAA	GCCCCAGACCTGGTGAATTA
ENSOART00020040698	GGTCCTCTCCCTCTGTTGAC	AAATACCGCCCATCTCCACC
MSTRG.11461.1	CGGTGTCTCACTGGTAGCTC	ACTCTCGCCTTCGTTCACAG
MSTRG.4748.1	GAGCCCGGAACCCGAAATAG	CGTATCCAACAGTGCCTCGT
MSTRG.12379.1	AGGAACCAACGTACCTGTCTC	TGCTTCCTGGTCCTATAGCAGT
MSTRG.14712.2	AGGGGGAGTGTTAAGTGGGT	GATCCAACAACCCCACAGGA
ENSOART00020036371	CACTGCTACCCGTTGAGGAA	TTCCCCGGTTGCATTCTGTC
ENSOART00020006750	CTTTGTGGTTCTCCCCGTTTTC	GTCTCTGCTGTGGGTCTCCTG
ENSOART00020037755	CCGACCTCAGTGAAGGGAAC	CCTCTGCTCCAAGACTGACG
MSTRG.13274.2	GGGGCATCTGAGAACAACCA	AAACGAGTGCAGCTTTGCAG
β-actin	AGCCTTCCTTCCTGGGCATGGA	GGACAGCACCGTGTTGGCGTAA
GAPDH	GTCGGAGTGAACGGATTTGG	ACGATGTCCACTTTGCCAGT

## Data Availability

The RNA-Seqs data have been deposited to the National Center for Biotechnology Information (https://dataview.ncbi.nlm.nih.gov/object/PRJNA1039176?reviewer=bl16hv174uhdqjb3iuiuct22eo, upload on 11 November 2023), with the BioProject accession PRJNA1039176. Proteome data have been deposited to the iProX (https://proteomecentral.proteomexchange.org/cgi/GetDataset?ID=PXD047077, upload on 20 November 2023), with the PXD accession PXD047077.

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
