# Peer review of "LncRNA and Protein Expression Profiles Reveal Heart Adaptation to High-Altitude Hypoxia in Tibetan Sheep"

_ijms, 2023, doi:10.3390/ijms25010385_

Round 1
Reviewer 1 Report
Comments and Suggestions for Authors
He et al present a fascinating study regarding lncRNA in adaption to high altitude.
The manuscript is well-written an follows a logical sequence, with each study section justifiably leading into the next.
One aspect that always stands out in studies like this, is whether the changes observed are a ‘cause or effect’ of the treatment (or exposure, in this instance). In this manuscript, the authors do a reasonable job of justifying the observed changes as being responsible for altitude acclimation.
Nonetheless, one concern, at least in terms of data interpretation, is the 12hr fasting prior to tissue harvest. My concern regarding fasting, is that many of the identified genes and pathways (eg APOA4, PPAR) respond to fasting. And while there may be altitude-dependent changes in expression of these genes, are they simply altitude-dependent changes, or altitude-dependent changes to the fasting response? This may require some discussion or noting.
Similarly, were the ewes slaughtered at their respective altitudes, or elsewhere? If elsewhere, how long between being removed from altitude to slaughter?
Method of slaughter should probably be noted in the methods.
Figure 5 – what does the red/grey shading represent? It is somewhat outlined in the results, at least for 5A, but should also be in the figure legend.
Figure 9 figure legend appears to be replication of Figure 8 figure legend.
Was the expression level of selected genes/proteins determined (APOA4, APOC3, 265 APOA2, APOH, ALDOB, FBP1, ADH1C, FABP4)?
There appears to be some data regarding this in the discussion, but this would arguably be well presented in the results section.
Discussion outlining ALDOB reports that expression at TS45 is lower than that at TS35. Why would the sheep adapt at 4500m but not at 3500?
While perhaps unachievable at this stage, comparison to a ‘plains’ equivalent sheep would have been very beneficial.
Author Response
Dear Reviewer:
Thank you for your review and suggestions on our manuscript entitled “Lnc RNA and Protein expression profiles reveal Heart Adaptation to High-Altitude Hypoxia in Tibetan Sheep” (ID: ijms-2769602). Those comments are all valuable and very helpful for revising and improving our paper. We have studied comments carefully and have made correction which we hope meet with approval.
The main corrections in the paper and the responds to your comments are as flowing:
Point 1: Nonetheless, one concern, at least in terms of data interpretation, is the 12hr fasting prior to tissue harvest. My concern regarding fasting, is that many of the identified genes and pathways (eg APOA4, PPAR) respond to fasting. And while there may be altitude-dependent changes in expression of these genes, are they simply altitude-dependent changes, or altitude-dependent changes to the fasting response? This may require some discussion or noting.
Response: Thank you again for carefully reading our manuscript and giving us valuable suggestion. In this study, we ensured that all animals at three altitudes underwent a 12-hour fasting period before slaughter, resulting in a relatively consistent impact of fasting on them. Consequently, our discussion primarily focuses on elucidating and examining the altitude-related factors that account for the variations in gene expression levels.
Point 2: Similarly, were the ewes slaughtered at their respective altitudes, or elsewhere? If elsewhere, how long between being removed from altitude to slaughter?
Response: Thank you for your valuable suggestion, which holds significant importance. In this study, animals from each of the three groups were slaughtered at their respective altitudes, and we added the content in “4.2. Animals and sample collection”. (Line 415)
Point 3: Method of slaughter should probably be noted in the methods.
Response: The detail was added in “4.2. Animals and sample collection”. All ewes were slaughtered in a humane manner, adhering to Islamic practices, which involved exsanguination, peeling, and splitting down the midline following standard operating procedures. The experiment was approved by the Animal Care Committee at Gansu Agricultural University, with due consideration given to animal welfare and conditions during the use of experimental animals. (Line: 415-420)
Point 4: Figure 5 – what does the red/grey shading represent? It is somewhat outlined in the results, at least for 5A, but should also be in the figure legend.
Response: Thank you again for carefully reading our manuscript and giving us valuable suggestion. In Figure 5, the red/grey shading indicates the significantly enriched profiles of DE lncRNAs or DEPs identified in the trend analysis. We have also made further clarifications in the manuscripts. (Line: 209-210 and line: 221-222)
Point 5: Figure 9 figure legend appears to be replication of Figure 8 figure legend.
Response: Thank you for your valuable suggestion. We have modified in the manuscript. (Line: 286-288)
Line 286-288: Figure 9. Correlation analysis. (A) Sankey diagram of KEGG pathways and GO terms for DE lncRNA target genes of adaptability to Highland Hypoxia; (B) Pearson correlation analysis between genes and blood physiological and biochemical indexes.
Point 6: Was the expression level of selected genes/proteins determined (APOA4, APOC3, 265 APOA2, APOH, ALDOB, FBP1, ADH1C, FABP4)?
There appears to be some data regarding this in the discussion, but this would arguably be well presented in the results section.
Response: Thank you again for carefully reading our manuscript and giving us valuable suggestion. In the discussion section, we mentioned the changes in the expression of some common differential genes/proteins (e.g., ALDOB protein) related to hypoxia adaptation in Tibetan sheep, based on the expression levels of corresponding differential genes/proteins in the heart transcriptome/proteome sequencing results of Tibetan sheep at different altitudes in the present study, to further analyse and explore the mechanisms involved. (Line 371-374, line: 390-403).
Point 7: Discussion outlining ALDOB reports that expression at TS45 is lower than that at TS35. Why would the sheep adapt at 4500m but not at 3500?
Response: This section is based on the current findings of ALDOB protein expression in proteomic sequencing, aiming to speculate on the potential cause. Regarding your mention of the higher ALDOB protein expression level in Tibetan sheep at an altitude of 3500m compared to the other two altitudes (2500m and 4500m), I believe it may not be due to the Tibetan sheep at 3500m not adapting to high-altitude low oxygen environments. Instead, this could be attributed to varying adaptive mechanisms that Tibetan sheep have developed to cope with the low oxygen environment at different altitudes.
As discussed, excessive vascular reconstruction could lead to arteriosclerosis, resulting in the loss of proper size adjustment and eventually leading to atherosclerosis in humans and animals [58,59] (Line:369-371). In comparison to Tibetan sheep at 2500m, Tibetan sheep at 3500m might have undergone some vascular remodeling, but it remains within the acceptable range for the animal body and does not cause arterial wall sclerosis, and the impact of vascular remodeling on Tibetan sheep at 3500m may not be as significant as at higher altitudes, such as 4500m. Therefore, the expression level of ALDOB protein in the heart of Tibetan sheep is relatively high at 3500m.
These are my personal insights and inferences on this matter, which may not be entirely accurate. However, based on the existing findings, I have attempted to explain the biological issues as comprehensively as possible. In future research, we will further investigate the underlying reasons for this phenomenon.
Once again, thank you for your valuable feedback. Your assistance has greatly improved our manuscript.
Point 8: While perhaps unachievable at this stage, comparison to a ‘plains’ equivalent sheep would have been very beneficial.
Response: Thank you for your suggestion. In this study, we want to study the hypoxia adaptability of Tibetan sheep, an indigenous livestock on the plateau. The comparison between plateau and plain varieties is also a very meaningful study. In future research, we will design the experiment as you suggested.
Sincerely,
Zhaohua He
Reference:
- Humphrey, J.D. Mechanisms of Vascular Remodeling in Hypertension. Am J Hypertens 2021, 34, 432-441.
- Tanaka, L.Y.; Laurindo, F.R.M. Vascular remodeling: A redox-modulated mechanism of vessel caliber regulation. Free Radical Bio Med 2017, 109, 11-21.

Reviewer 2 Report
Comments and Suggestions for Authors
incredibly interesting work, the results of which may become the direction of therapeutic treatment in the future. Interesting introduction, methodology well described, results well presented and well discussed.
I suggest that you take my two suggestions into account in your work:
- in the introduction, it is worth adding some information (in a few sentences) about pharmacological methods of preventing altitude sickness (provide the mechanisms used). I think it will allow readers to better understand the importance of the published results.
- the figures are rich in content and quite small. The magazine's concept allows for publishing figures either in a strip of text or in the full width of the page. I suggest choosing the second option and enlarging the drawings as much as possible. The reader will then avoid the need to constantly use a magnifying glass, apart from the fact that in the printed version the readability of the figures is poor.
Author Response
Dear Reviewer:
Thank you for your review and support on our manuscript entitled “Lnc RNA and Protein expression profiles reveal Heart Adaptation to High-Altitude Hypoxia in Tibetan Sheep” (ID: ijms-2769602). In future scientific research endeavors, we will persistently endeavor to attain increasingly valuable outcomes in our scientific investigations.
The main corrections in the paper and the responds to your comments are as flowing:
Point 1: - in the introduction, it is worth adding some information (in a few sentences) about pharmacological methods of preventing altitude sickness (provide the mechanisms used). I think it will allow readers to better understand the importance of the published results.
Response: Thank you again for carefully reading our manuscript and giving us valuable suggestion. We have added the appropriate content to the manuscript. (Line: 46-53)
Line 46-53: The mechanisms of altitude sickness are complex. Hypoxia is the cause, while upper respiratory tract infections, fatigue, cold, stress, hunger, pregnancy and other factors are all causative factors [8]. Hypoxia triggers the consistent activation of HIF, which in turn triggers the activation of numerous hypoxia-related genes, including EPO, which facilitates the production of red blood cells. This process becomes excessively active during acute hypoxic exposure and can result in abnormal erythropoiesis. Consequently, the elevated number of red blood cells increases blood viscosity, predisposes individuals to pulmonary hypertension, and causes damage to the microcirculation [3,9].
Point 2: - the figures are rich in content and quite small. The magazine's concept allows for publishing figures either in a strip of text or in the full width of the page. I suggest choosing the second option and enlarging the drawings as much as possible. The reader will then avoid the need to constantly use a magnifying glass, apart from the fact that in the printed version the readability of the figures is poor.
Response: Thank you again for carefully reading our manuscript and giving us valuable suggestion. I will submit the original Figures from the manuscript to the editorial office. In the event that our manuscript gets accepted and we have the opportunity to participate in the proofreading stage, we will collaborate with the editorial team. I If they agree, the images will be placed on the entire page after the document during the proofreading process.
Reference:
- West, J.B. Physiological Effects of Chronic Hypoxia. New Engl J Med 2017, 376, 1965-1971.
- Persson, P.B.; Bondke Persson, A. Altitude sickness and altitude adaptation. Acta Physiol. 2017, 220, 303-306.
- Beall, C.M. Adaptation to High Altitude: Phenotypes and Genotypes. Annual Review of Anthropology 2014, 43, 251-272.
